# Persistent Human KIT Receptor Signaling Disposes Murine Placenta to Premature Differentiation Resulting in Severely Disrupted Placental Structure and Functionality

**DOI:** 10.3390/ijms21155503

**Published:** 2020-07-31

**Authors:** Franziska Kaiser, Julia Hartweg, Selina Jansky, Natalie Pelusi, Caroline Kubaczka, Neha Sharma, Dominik Nitsche, Jan Langkabel, Hubert Schorle

**Affiliations:** 1Department of Developmental Pathology, Institute of Pathology, University Hospital Bonn, 53127 Bonn, Germany; Franziska.Kaiser@ukbonn.de (F.K.); E_Hartweg_J@ukw.de (J.H.); s.jansky@kitz-heidelberg.de (S.J.); Natalie.Pelusi@ukbonn.de (N.P.); Caroline.Schuster-Kubaczka@childrens.harvard.edu (C.K.); ns.nehasharma01@gmail.com (N.S.); s6donits@uni-bonn.de (D.N.); Jan.Langkabel@ukbonn.de (J.L.); 2Department of Medicine II and IZKF Research Laboratory, Würzburg University Hospital, 97080 Würzburg, Germany; 3Hopp Children’s Cancer Center (KiTZ), 69120 Heidelberg, Germany; 4Division of Neuroblastoma Genomics, German Cancer Research Center (DKFZ), 69120 Heidelberg, Germany; 5Department of Molecular Pathology, Institute of Pathology, University Hospital Bonn, 53127 Bonn, Germany; 6Division of Pediatric Hematology/Oncology, Children’s Hospital Boston, Harvard Medical School, Boston, MA 02115, USA; 7Department of Obstetrics and Gynaecology, Yong Loo Lin School of Medicine, National University of Singapore, Singapore 119077, Singapore; 8Life & Medical Sciences-Institute (LIMES), University of Bonn, 53115 Bonn, Germany

**Keywords:** KIT receptor, KITD816V, placental development, premature differentiation, trophoblast stem cell, trophoblast giant cell, spongiotrophoblast, invasion, embryonic growth retardation

## Abstract

Activating mutations in the human KIT receptor is known to drive severe hematopoietic disorders and tumor formation spanning various entities. The most common mutation is the substitution of aspartic acid at position 816 to valine (D816V), rendering the receptor constitutively active independent of ligand binding. As the role of the KIT receptor in placental signaling cascades is poorly understood, we analyzed the impact of KIT^D816V^ expression on placental development using a humanized mouse model. Placentas from KIT^D816V^ animals present with a grossly changed morphology, displaying a reduction in labyrinth and spongiotrophoblast layer and an increase in the Parietal Trophoblast Giant Cell (P-TGC) layer. Elevated differentiation to P-TGCs was accompanied with reduced differentiation to other Trophoblast Giant Cell (TGC) subtypes and by severe decrease in proliferation. The embryos display growth retardation and die in utero. KIT^D816V^-trophoblast stem cells (TSC) differentiate much faster compared to wild type (WT) controls. In undifferentiated KIT^D816V^-TSCs, levels of Phosphorylated Extracellular-signal Regulated Kinase (P-ERK) and Phosphorylated Protein Kinase B (P-AKT) are comparable to wildtype cultures differentiating for 3–6 days. Accordingly, P-TGC markers Placental Lactogen 1 (PL1) and Proliferin (PLF) are upregulated as well. The results reveal that KIT signaling orchestrates the fine-tuned differentiation of the placenta, with special emphasis on P-TGC differentiation. Appropriate control of KIT receptor action is therefore essential for placental development and nourishment of the embryo.

## 1. Introduction

Proper function of the placenta is essential for the development of the embryo as it is responsible for exchanging gases, nutrients, and waste products between the mother and the fetus. Placental insufficiency can result in adverse effects for the embryo including intrauterine growth retardation, embryonic defects, or even fatal outcome. Therefore, placenta development and differentiation are closely regulated [1,2,3].

The membrane-bound tyrosine receptor kinase KIT is activated by its ligand stem cell factor (SCF), which causes dimerization of the receptor followed by phosphorylation of its tyrosine residues [4,5,6]. Activation of KIT results in induction of various downstream signaling cascades such as the Mitogen-Activated Protein Kinase (MAPK)/Extracellular-Signal Regulated Kinase (ERK) and Janus Kinase (JAK)/Signal Transducers and Activators of Transcription (STAT) pathways which orchestrate cell proliferation, angiogenesis, cell migration, and cell cycle control [7,8,9,10]. During placental development, the KIT protein is detected in the uterine epithelium as well as the maternal and fetal part of the placenta. Importantly, KIT expression has not been detected in uteri of nonpregnant females, indicating a pregnancy-related role of KIT signaling [11,12,13]. In detail, starting at E9, KIT and SCF are detectable in trophoblast-chorion, ectoplacental cone, and decidua [12]. Placental hematopoietic activity begins around mid-gestation. Hematopoietic stem cells (HSC) are detected in the placenta at E11, peak in numbers at E12 to E13, and disappear thereafter [13,14]. These HSC were shown to be KIT positive, whereas endothelial cells surrounding the HSC niches are SCF positive [13,15]. At E12, KIT can also be found in mesenchymal cells of the chorionic plate and of the labyrinth, in endothelial cells lining the vessels, as well as in few trophoblast giant cells (TGC) [13]. At E14.5, KIT is mainly expressed in maternal decidua cells as well as the labyrinth whereas SCF is found only in labyrinth [11,12]. KIT is restricted to labyrinthine trophoblast cells exposed to maternal blood at E19 and is no longer detected in endothelial or mesenchymal cells [11].

The KIT receptor has been implicated with several disorders such as tumor formation, hematopoietic disorders, and systemic mastocytosis [16,17,18,19,20]. The most common activating mutation in the human KIT receptor encodes for the substitution of the aspartic acid at position 816 with a valine (D816V) causing constitutive phosphorylation of the protein [17,21,22]. Depletion of KIT receptor results in various defects spanning from hematopoietic failure, macrocytic anemia, pigmentation deficiency, and sterility to intestinal dysfunction [23,24,25,26]. Mice carrying homozygous KIT^W/W^ mutation die pre- or perinatally; however, can be rescued by microinjection of wildtype fetal liver hematopoietic cells into placentas of affected fetuses [23,24,26]. Introduction of the viable c-Kit-deficient mouse line allowed for studying the loss of KIT expression in the adult mouse and showed that KIT is essential for adult lymphopoiesis in bone marrow and thymus [26]

Previously, we reported the generation of a humanized KIT^D816V^ mouse model [27]. The transgenic KIT receptor consists of murine extracellular and transmembrane domains and the human intracellular domain carrying the D816V mutation. The chimeric KIT receptor is fused to a green fluorescent protein (GFP) and integrated in the ROSA26 locus, allowing for Cre-mediated conditional expression of the protein under control of the endogenous ROSA26 promoter [27,28]. We then examined KIT signaling in fetal liver erythropoiesis and adult hematopoiesis [27,29]. Constitutively active KIT resulted in inhibition of terminal differentiation of erythroid precursors and embryonic death after embryonic day (E) 13.5 [27]. In adult mice, expression of KIT^D816V^ led to a polycythemia vera-like myeloproliferative neoplasm with highly increased red blood cells and splenomegaly [29].

While the role of KIT signaling in other tissues is well understood, its role in placental development, however, has so far not been described. Here, we examine the effect of the constitutively active KIT receptor using the KIT^D816V^ mouse model. Interestingly, expression of KIT^D816V^ resulted in decreased proliferation of trophoblast cells. As a consequence, placental structure was affected. The labyrinth layer was decreased accompanied by increased differentiation to parietal (P-)TGCs, whereas other TGC subtypes remained underrepresented. Trophoblast Stem Cells (TSC) derived from KIT^D816V^ blastocysts showed altered activation of signaling pathways upon differentiation as well as an increased invasive capacity. Due to these effects, the placenta is not able to sustain regular development and the embryos show severe growth retardation and die in utero.

## 2. Results

### 2.1. Embryos and Placentas Carrying KIT^D816V^ Mutation Suffer from Severe Growth Retardation

Previously, we generated the R26-LSL-KIT^D816V^ mouse line, which allows for conditional Cre-induced expression of chimeric KIT^D816V^ receptor and a GFP reporter driven by the ROSA26 promoter. The KIT^D816V^ cDNA is linked to the GFP cDNA via the coding sequence for a viral 2A peptide [27]. Here, mice carrying the ROSA26-KIT^D816V^-GFP transgene were mated with Deleter-Cre mice, inducing ubiquitous expression of the transgenic receptor in the embryonic as well as the extra-embryonic tissue. Presence of ROSA26-KIT^D816V^ and Cre transgenes was verified by genotyping PCR using genomic DNA obtained from yolk sac or embryo (Appendix A). Animals/placentas harboring both the Cre- and the ROSA26-KIT^D816V^ allele are further referred to as KIT^D816V^ animals/placentas. Presence of transgenes in KIT^D816V^ animals was further validated by analyzing RNA expression and protein levels in KIT^D816V^ placentas (Figure 1A–C). Of note, in KIT^D816V^ animals expressing human *KIT^D816V^*, the expression of murine *Kit* remained unchanged, showing that endogenous *Kit* expression is not affected by transgene induction (Figure 1A,B). As expected, KIT as well as 2A-peptide proteins were present in KIT^D816V^ placentas only (Figure 1C). For further analyses, embryos and placentas were obtained on E9.5–E11.5 from KIT^D816V^ and wildtype (WT) animals. KIT^D816V^ embryos and placentas dissected on E11.5 showed growth retardation and developmental delay (Figure 1D). We previously reported that KIT^D816V^ expression restricted to the embryo proper leads to disturbance of the hematopoietic system and that such animals die at E14.5. Therefore, we hypothesize that severe growth retardation observed in KIT^D816V^ animals at 11.5 is an effect of placental insufficiency. Of note, KIT^D816V^ embryo and placentas showed GFP positivity (Figure 1D).

### 2.2. KIT^D816V^ Placentas Show Fewer Proliferating Cells

Next, we examined whether reduced sizes in KIT^D816V^ placentas occurred due to diminished proliferation. Proliferation in placental tissue was assessed by immunohistochemical staining for KI-67—a protein that is present in proliferating cells as well as cells undergoing endoreduplication [30,31]. At both timepoints analyzed, KI-67-positive cells were mainly detectable in the labyrinth and spongiotrophoblast layers of KIT^D816V^ and WT placentas. On day E9.5, the labyrinth layer and overall amount of KI-67-positive cells were markedly diminished in KIT^D816V^ in comparison to WT (Figure 2A,B). While the labyrinth and spongiotrophoblast layers were enlarged on day E10.5 in comparison to E9.5 in both samples, less KI-67-positive cells were detected in KIT^D816V^ placentas than in WT placentas (Figure 2A,B). KI-67-positive P-TGCs can be detected in both KIT^D816V^ and WT placentas. These results suggest that the smaller size of KIT^D816V^ placentas results from decreased proliferation in the labyrinth and spongiotrophoblast.

### 2.3. KIT^D816V^ Placentas Show Reduced Labyrinth Layer and Disrupted Formation of Vasculature

To assess the placental structure, hematoxylin and eosin (H&E) staining was performed on paraffin sections of KIT^D816V^ and WT placentas. Starting from day E9.5 in KIT^D816V^ placentas, H&E stained sections and quantification of the area showed a reduction in labyrinth size (Figure 3A,B). Also, the spongiotrophoblast layer was slightly reduced (Figure 3A,B). Interestingly, at E10.5, the layer of P-TGCs was increased in KIT^D816V^ placentas (Figure 3A,B). Quantification of P-TGCs within this area confirmed a significantly higher total number of P-TGCs in KIT^D816V^ sample (Figure 3C). Next, RNA was isolated from KIT^D816V^ and WT placentas on days E10.5 and E11.5. Expression of labyrinth TGC markers Placental Lactogen 2 (*Pl2)*, Cathepsin q (*Ctsq)*, and Glial Cells Missing Homolog 1 (*Gcm1)* is diminished in comparison to WT placentas (Figure 3D). This data indicates that the number of *Pl2*^+^ and *Ctsq*^+^ sinusoidal (S-)TGC and the number of *Gcm 1*^+^ labyrinthine cells is reduced.

Labyrinthine architecture was further examined by performing anti-CD31 staining. CD31 is expressed in fetal endothelial cells [32]. The staining revealed that at E9.5 there are fewer CD31-positive cells present in KIT^D816V^ placentas in contrast to WT placentas (Figure 3E). CD31-positive cells are indicated by red arrowheads. While the vasculature was established from E9.5 to E10.5 in KIT^D816V^ as well as WT placentas, the capillary network in KIT^D816V^ remained less prominent than in WT tissue. S-TGCs, indicated by black arrowheads, were lining maternal sinusoids (red asterisks) in both conditions. Surprisingly, maternal blood was detected in between P-TGCs of KIT^D816V^ placentas at E10.5, suggesting a disruption of the developing vascular structure (Figure 3E).

### 2.4. KIT^D816V^ Placentas Show Prominent Differentiation into P-TGCs

Since the number of P-TGCs is significantly increased in KIT^D816V^ placentas (Figure 3C), we next investigated this TGC subtype in more detail by detecting the P-TGC markers PL1 and PLF using in situ hybridization. At E9.5, the number and distribution of cells positive for PL1 and PLF appeared unaltered in KIT^D816V^ compared to WT placentas (Figure 4A). At E10.5, however, the number of PL1^+^/PLF^+^ cells was visibly higher in KIT^D816V^ placentas than WT placentas (Figure 4B). Further, when comparing PL1 and PLF staining in WT placentas, we observed areas that are PL1 negative but PLF positive (Figure 4B). Such cells are not P-TGCs but rather invading Spiral Artery (SpA-)TGCs (black arrowhead). We assume that the cell clusters indicated by red arrowheads are canal (C-) TGCs (Figure 4B). By contrast, PL1^−^/PLF^+^ cells are not detected in KIT^D816V^ placentas, further suggesting a severe reduction in specific TGC subtypes such as SpA-TGC and C-TGC.

### 2.5. Spongiotrophoblast Cells and Glycogen Trophoblasts are Reduced in KIT^D816V^ Placentas

Next, we analyzed the development of the spongiotrophoblast layer in more detail. In situ hybridization revealed that the spongiotrophoblast marker Trophoblast Specific Protein Alpha (TPBPA) is hardly present in KIT^D816V^ placentas compared to WT placentas at both timepoints analyzed (Appendix A). Also, expression of *Tpbpa* was significantly reduced at E10.5 and E11.5. Further, we investigated the expression of Gap Junction Beta 3-protein (*Gjb3)* and Procadherin 12 (*Pcdh12)*, both markers for glycogen trophoblasts (GlyT). They were significantly decreased in KIT^D816V^ placentas on days E10.5 and E11.5 (Appendix A). Thus, these results suggest that expression of KIT^D816V^ not only blocks cellular proliferation but also heavily impinges on the fine-balanced differentiation of the various subtypes essential for placental proper development. The differentiation to P-TGCs seems increased at the expense of other TGC subtypes such as C-TGCs and SpA-TGCs as well as spongiotrophoblast and GlyT cells.

### 2.6. KIT^D816V^-TSC Show Signs of Premature Differentiation

In order to analyze the molecular effect of KIT^D816V^ on a cellular level in more detail, TSCs were derived from E3.5 blastocysts obtained from R26-LSL-KIT^D816V^ and Deleter-Cre mice according to published procedures [27,33]. Genotyping at passage five was used to distinguish between KIT^D816V^- and WT-TSC lines (Appendix A). Ultimately, we had established two lines of KIT^D816V^-TSC which were named KIT^D816V^ #3 and KIT^D816V^ #4. In these two lines, we were able to detect a GFP-signal using fluorescence-activated cell sorting (FACS) (Appendix A) and expression of the human *KIT* transgene using qRT-PCR (Appendix A), demonstrating that the ROSA26-GFP-2A-KIT^D816^ allele is functional in TSC culture in vitro. As in placental tissues, the level of endogenous murine *Kit* expression in KIT^D816V^-TSC is not affected and remains comparable to that of WT TSC (Appendix A).

Analysis of TSC-markers Transcription Factor AP-2 Gamma (Tfap2c), Caudal Type Homeobox 2 (Cdx2), and Eomesodermin homolog (Eomes) revealed that neither expression (Appendix A) nor protein levels and distribution of TFAP2C, CDX2, and EOMES (Appendix A) are affected in the KIT^D816V^-TSC lines. Also, morphology as well as splitting ratio and frequency of KIT^D816V^-TSC did not differ from WT TSC. Hence, we conclude that establishment, maintenance, and self-renewal of TSC is not altered by expression of the KIT^D816V^ transgene.

To evaluate whether KIT^D816V^-TSC displays alterations in differentiation in vitro, WT and KIT^D816V^ lines were kept under differentiating conditions in trophoblast stem (TS) cell medium without conditioned medium (CM), Fibroblast Growth Factor (FGF) 4, and heparin for 9 days. Morphological analyses show no difference between studied lines. In both KIT^D816V^-TSC and WT TSC, TGCs appeared in culture after three days (Figure 5A). RNA was isolated on days 0 and 6 and was analyzed for expression of P-TGC markers *Pl1*, *Pl2*, and *Plf*. Interestingly, in KIT^D816V^-TSCs, all markers showed an increased level already at day 0, an effect which persisted to day 6 (Figure 5B). Further, spongiotrophoblast marker *Tpbpa* as well as labyrinth and S-TGC marker *Ctsq* and GlyT markers *Gjb3* and *Pcdh12* were expressed at lower levels in KIT^D816V^ TSC than in WT TSC (Appendix A). Upregulation of *Tfap2c* which is detected during TSC differentiation was higher in KIT^D816V^-TSC than in WT TSC [34] (Appendix A). Of note, *Tfap2c* expression was already increased under stem cell culture conditions in KIT^D816V^-TSC. *Hand1* is involved in mediation of TGC differentiation and is expressed in the ectoplacental cone [35]. In KIT^D816V^-TSC it is significantly upregulated in comparison to WT-TSC after 6 days of differentiation (Appendix A). Finally, we and others had previously shown that *Gata2* expression was upregulated upon KIT signaling in cells of the hematopoietic system [27,36]. Here, in KIT^D816V^ TSC, we detected a significant increase of *Gata2* expression after six days of differentiation whereas *Gata2* expression levels remained constant in WT TSC (Appendix A). These results demonstrate that expression of spongiotrophoblast, labyrinth, and glycogen trophoblasts is underrepresented in KIT^D816V^ TSC upon differentiation consistent with data obtained from in vivo samples. Also, expression of KIT^D816V^ leads to an upregulation of P-TGC markers and *Tfap2c* already existing in TSC culture, which leads to premature and skewed induction of differentiation of TSCs, which is not morphologically apparent.

### 2.7. Signaling Cascades and Invasion Capability are Affected in KIT^D816V^-TSC

Previously, we had shown that, in the hematopoietic system, expression of KIT^D816V^ leads to a block of differentiation of erythroblasts and a continuation of proliferation of precursor cells. There, we had demonstrated that KIT^D816V^ leads to induction of MAPK signaling as well as diminished AKT activation [27]. Hence, we used KIT^D816V^-TSC to examine the activation of key signaling pathways during 9 days of TSC differentiation (Figure 5C). On day 0, levels of phosphorylated (P-) ERK and P-AKT in KIT^D816V^ TSC appeared reduced and comparable to day 3/6 of differentiating WT-TSC. Of note, on day 0, levels of P-STAT3 are comparable but become upregulated over the course of differentiation in KIT^D816V^-TSC (Figure 5C). This suggests that chronic activation of the KIT signaling cascade in KIT^D816V^-TSC results in diminished levels of P-ERK and P-AKT, priming the cells for rapid and premature differentiation.

Among their various functions such as guiding the attachment of the blastocyst and secretion of essential hormones and proteins, TGCs are also capable of invading into uterine tissue to establish the vital connection to the maternal blood vessels [31,37]. Finally, we analyzed the invasive capacity of KIT^D816V^-TGCs in comparison to WT-TGCs by using a Transwell migration assay. Previously, it was shown that higher Matrigel concentrations result in preselection of TGCs as non-TGCs do not invade through thicker Matrigel layers [38]. Also, lower cell densities led to increased invasion [38]. Hence, we seeded 2 × 10^4^ cells on a layer of 0.8 mg/mL Matrigel and omitted FGF4, heparin, and CM from culture medium to induce differentiation. After five days, Hoechst staining showed the presence of nuclei resembling TGCs in size and structure, indicating successful invasion (Figure 5D). Quantification of two independent experiments revealed that a significantly higher number of KIT^D816V^-TGCs had migrated through the Matrigel and the pores (Figure 5E). Thus, KIT^D816V^ signaling in trophoblast cells seems to result in a higher portion of invasive cells. This might be due to the fact that KIT^D816V^-TSC are much faster in inducing differentiation and gain migratory capabilities earlier than WT-TSC.

## 3. Discussion

Here, we show several effects of KIT^D816V^ on murine placental development. These findings comprise aberrant placental structure and increased differentiation into P-TGC subtypes, whereas other TGC subtypes such as SpA-TGCs, S-TGC, and C-TGCs remain underrepresented. Since TGC variety is essential for placental development, diminished levels of SpA-TGCs may lead to impaired establishment of blood flow at the implantation site by not properly formed blood vessels, lack of dilating spiral arteries, and insufficient secretion of vasodilators and other angiogenic factors [31,37]. C-TGCs contribute to formation of the vessels delivering maternal blood to the labyrinth [39]. Loss of these highly specialized cell types affects invasion, exchange of nutrients and gases, as well as hormone secretion, resulting in decreased functionality of the placenta. Moreover, proliferation in KIT^D816V^ placentas ceases at E10.5 when KI-67-positive cells cannot be detected anymore. At that timepoint, the embryo already shows severe growth retardation. Although we cannot exclude additive embryonic effects due to KIT^D816V^ expression, we suspect this phenotype to result from malnourishment due to placental failure.

While in other cell types activation of KIT resulted in induction of proliferation and impairment of differentiation [40], trophoblast and placenta development are hallmarked by reduced proliferation in combination with increased and skewed differentiation. In cells of the hematopoietic system, it had previously been shown that KIT signaling results in upregulation of c-Myc, c-Myb, and Gata2 [27,36]. Interestingly, in trophoblast cells, Gata2 binds to and transactivates expression of Pl1. Further, Gata2 level was also shown to be correlated with Plf expression [41,42]. In this study, KIT^D816V^-TSC showed a significant increase in *Gata2* expression upon differentiation. Therefore, we speculate that, also in trophoblast cells, KIT signaling induces Gata2 which in turn transactivates Pl1 and Plf. This would lead to a rapid induction of differentiation, which is accompanied by exiting from the cell cycle. It also explains the high number of PL1^+^/PLF^+^ cells observed in KIT^D816V^ placentas. In the context of premature differentiation, it is interesting to note that we were able to generate self-renewing TSC lines. The established TSC did not show any aberrant overall growth parameters and ease of handling. However, the fact that, at day 0 of differentiation, P-ERK and P-AKT levels were comparable to day 3 of regular differentiation together with the already upregulated markers for TGC-differentiation strongly suggests that the growth conditions (FGF4, Heparin, and CM) are able to override the KIT-mediated signals leading to differentiation. This helps to explain the phenotype observed in KIT^D816V^-placentas. During pre- and early post-implantation development, trophectoderm and ectoplacental cone cells rely on FGF4-induced signaling cascades leading to expression of trophectoderm and TSC master regulators Cdx2, Tfap2c, and Elf5 [43,44,45]. This protects the cells from differentiation inducing the effect of KIT signaling. Post implantation, when the different layers of the placenta are laid down, FGF signaling and expression of the marker genes decline. Since the KIT signaling cascade is already in place and active (as hallmarked by upregulation of *Pl1* and *Plf* at day 0 of TSC differentiation), a premature differentiation is induced, resulting in a smaller and disorganized placenta.

Interestingly, the placental alterations reported here are phenocopied in mouse models of Suppressor of Cytokine Signaling (SOCS) 3 deficiency as well as administration of retinoic acid (RA) to the mother. Administration of RA was demonstrated to result in loss of proliferation, differentiation to TGCs, and a reduced spongiotrophoblast layer [46]. Also, TPBPA levels were reduced whereas PL1 levels were increased [46]. Interestingly, in another study, RA resulted in an increase of invading TGCs [38]. In our model, chronic KIT activation results in significantly more invasive cells. Of note, RA was shown to promote KIT expression and translation in spermatogonia [47,48,49]. Hence, we speculate that, in trophoblast, KIT acts downstream of RA. In KIT^D816V^ mice, RA signaling is active, independent of RA presence (Figure 6).

In Leukemia Inhibitory Factor (LIF)/JAK/STAT3 signaling, Leukemia Inhibitory Factor (LIF) binds to its receptor and Janus Kinases (JAK) are activated. JAK then phosphorylates Signal Transducers and Activators of Transcription (STAT) 3 in the cytoplasm. Activated STAT3 induces the expression of SOCS3, which then represses LIF in a negative feedback loop [50]. Augmentation in LIF levels due to SOCS3 deficiency increases TGC differentiation [51]. SOCS3 deficiency was also reported to result in erythrocytosis and reduced spongiotrophoblast layer. Further, constitutively active STAT3 was observed [52]. We find increased levels of phosphorylated STAT3 in KIT^D816V^-TSC upon differentiation for 9 days in comparison to WT-TSC. STAT3 also plays a role in cell migration and invasion [53], both of which we demonstrate to be affected in the KIT^D816V^ mouse model. Taken together, we conclude that KIT is also involved in LIF/JAK/STAT3 signaling (Figure 6).

Here, we demonstrate that KIT^D816V^ placentas are severely affected by constitutively active KIT signaling. The results show that, in placenta development, KIT signaling is required for the fine-tuned induction of differentiation. Moreover, we speculate that KIT signaling is essential for P-TGC differentiation since this is the major TGC-type observed in the studies.

## 4. Materials and Methods

### 4.1. Generation of Transgenic Animals

All experiments were conducted according to the German law of animal protection and in agreement with the approval of the local institutional animal care committees (Landesamt für Natur, Umwelt und Verbraucherschutz, North Rhine-Westphalia, approval number: 84-02.03.2013/A428 approved date: 31 January 2014). R26-LSL-KIT^D816V^ mice described by us were mated with Deleter-Cre mice carrying human cytomegalovirus minimal promoter (CMV) controlled Cre recombinase, thereby allowing for ubiquitous expression of the mutant human KIT receptor [27,54]. Both mouse strains were kept on a 129sv/S2 and C57BL/6 genetic background. The morning after mating, female mice were checked for vaginal plugs. In the case of plug detection, noon of that day was considered E0.5. Accordingly, mice were sacrificed 9 to 11 days later around noon.

### 4.2. DNA Isolation and Genotyping PCR

Embryos and placentas were lysed in tissue lysis buffer (50 mM Tris-HCl (Roth, Karlsruhe, Baden-Wuerttemberg, Germany; #9090.1), 100 mM ethylenediaminetetraacetic acid (EDTA) (AppliChem, Darmstadt, Hessen, Germany; #A4975), 100 mM NaCl (AppliChem, Darmstadt, Hessen, Germany; #A1149), 1% (*w*/*v*) SDS (Merck, Darmstadt, Hessen, Germany; #817039), and 10 mg/mL proteinase K (Merck, Darmstadt, Hessen, Germany; #1245680500)), while DNA from adherent cells was obtained using cell lysis buffer (10 mM NaCl (AppliChem, Darmstadt, Hessen, Germany; #A1149), 10 mM Tris, 10 mM EDTA, 0.5% sarcosyl, and 1 mg/mL proteinase K (Merck, Darmstadt, Hessen, Germany)). DNA was precipitated using isopropanol, pelleted by centrifugation, and washed twice with ethanol (80%). The pellet was then resuspended in H_2_O (dd) at 37 °C. Using NanoDrop 1000 (Thermo Fisher Scientific, Waltham, MA, USA), concentration and purity of obtained DNA was measured. Genotyping was performed by polymerase chain reaction (PCR) using PCR primers. Primer sequences are listed in Appendix A.

### 4.3. RNA Analysis

RNA isolation from tissue and cells was carried out using TRIzol^TM^ reagent (Invitrogen, Thermo Scientific, Waltham, MA, USA; #15596026). RNA was precipitated using isopropanol and pelleted by centrifugation. After washing with ethanol (75%), pellet was resuspended in diethyl pyrocarbonate (DEPC)-treated water. Using NanoDrop 1000 (Thermo Fisher Scientific, Waltham, MA, USA), concentration and purity of obtained RNA was measured. DNase digestion (Thermo Scientific, Waltham, Massachusetts, USA; #EN0525) and cDNA synthesis (RevertAid Premium, Fermentas, Thermo Scientific, Waltham, MA, USA; #EP0441) were carried out on 1 μg RNA. Quantitative real time PCR (qPCR) was performed using SYBR Green Master Mix (Fermentas, Thermo Scientific, Waltham, MA, USA; #4309155) and ViiA7 (AppliedBiosystems, Life Technologies, Foster City, CA, USA). Primer sequences are listed in Appendix A. Target gene expression was normalized to the housekeeping reference gene *Gapdh*, reactions were performed in triplicate, and *p*-values were calculated using unpaired t-test.

### 4.4. Hematoxylin and Eosin Staining

Paraffin sections were rehydrated and incubated in hematoxylin (Roth, Karlsruhe, Baden-Wuerttemberg, Germany; #T865) for 3 min. Counterstaining with eosin (Roth, Karlsruhe, Baden-Wuerttemberg, Germany; #X883) was performed for 1 min. Sections were then dehydrated, incubated in xylene (VMP Chemie Kontor GmbH, #1000649172), and embedded with Entellan^®^ (Merck, Darmstadt, Hessen, Germany; #107960). Analysis of stained sections was performed with a DM LB microscope (Leica, Wetzlar, Hessen, Germany).

### 4.5. Western Blotting

Protein was obtained by lysing tissue or cells in radioimmunoprecipitation assay (RIPA) buffer (NEB, #9806). Protein concentration was measured using Pierce^TM^ BCA Protein Assay Kit (Thermo Scientific, Waltham, MA, USA; #23225) on an iMarkTM Microplate Reader (BioRad, Hercules, CA, USA). Proteins were electrophoresed in a polyacrylamide (10%) gel (Roth, Karlsruhe, Baden-Wuerttemberg, Germany; #3029.1) and transferred onto a Roti^®^ polyvinylidene fluoride (PVDF) membrane (Roth, Karlsruhe, Baden-Wuerttemberg, Germany; #T830.1). Membranes were blocked, stained with primary antibodies, and horseradish peroxidase (HRP)-coupled secondary antibodies. Antibodies and concentrations are given in Appendix A. Pierce^TM^ ECL Western Blotting Substrate (Thermo Scientific, Waltham, MA, USA; #32106) and ChemiDoc^TM^MP (Biorad, Hercules, CA, USA) were used for detection of stained proteins.

### 4.6. Immunohistochemical/Immunofluorescence Staining

For CD31 and KI-67 immunohistochemical staining, paraffin sections were incubated with antigen retrieval buffer (Medac GmbH, Wedel, Schleswig-Holstein, Germany; #PMB1-250) and blocked with peroxide block (Medac GmbH, Wedel, Schleswig-Holstein, Germany; #925B-05). After primary antibody staining, sections were incubated with HRP-coupled secondary antibody, followed by a 3,3′-diaminobenzidine (DAB, Medac GmbH, Wedel, Schleswig-Holstein, Germany; #957D-50) staining for 8 min. Nuclei were stained with hematoxylin (Roth, Karlsruhe, Baden-Wuerttemberg, Germany; #T865) For immunofluorescence staining, cells were fixed in formalin (4%, Merck, Darmstadt, Hessen, Germany; #100496), permeabilized using 0.5% (*v*/*v*) Triton X-100 (AppliChem, Darmstadt, Hessen, Germany; #142314), and blocked in 2% bovine serum albumin (BSA, Sigma-Aldrich, St. Louis, MO, USA) and 0.1% Triton X-100 in phosphate-buffered saline (PBS). Then, cells were stained with primary antibody and Alexa-Fluor-conjugated secondary antibodies. Nuclei were stained with Hoechst (Sigma-Aldrich, St. Louis, MO, USA; #H6024). Antibodies and concentrations are given in Appendix A.

### 4.7. In Situ Hybridization

In situ hybridization was performed using protocol and digoxigenin (DIG)-labeled probes established by Simmons et al. [37]. In short, cryo- or paraffin sections were thawed or deparaffinized and rehydrated, respectively. Sections were fixed using 4% paraformaldehyde (Merck, Darmstadt, Hessen, Germany; #818715) and treated with proteinase K (20 μg/mL, Merck, Darmstadt, Hessen, Germany; #1245680500). After another fixation step, sections were blocked. Probes (2 ng/μL) for PL1, PLF, and TPBPA were denatured and incubated with sections overnight, followed by a RNase A (AppliChem, Darmstadt, Hessen, Germany; #A2760) treatment. Sections were blocked, and anti-digoxigenin-AP (1:1000, Sigma-Aldrich, St. Louis, MO, USA; #11093274910) was added for probe detection. Incubation with BM-purple followed overnight; then, sections were counterstained with nuclear fast red (Sigma-Aldrich, St. Louis, MO, USA; #N3020). Sections were dehydrated, treated with xylene (VMP Chemie Kontor GmbH, Siegburg, North Rhine-Westfalia, Germany; #1000649172), and mounted in Entellan^®^ (Merck, Darmstadt, Hessen, Germany; #107960). Leica DM LB microscope (Wetzlar, Hessen, Germany) was used for analysis of sections.

### 4.8. Cell Culture

KIT^D816V^-TSCs were derived from E3.5 blastocysts, which were obtained after mating of R26-LSL-KIT^D816V^ mice with Deleter-Cre mice [33]. Cre recombinase expression is controlled by human CMV minimal promoter [54]. Cells were cultured on tissue culture plastic in humidified incubators (Heracell 240i, Thermo Scientific, Waltham, MA, USA) with 7.5% CO_2_ at 37 °C. TSC were grown in a trophoblast stem (TS) cell medium supplemented with 70% mouse embryonic fibroblast (MEF) conditioned medium (CM) [45]. TS medium was supplemented with 25 ng/mL human recombinant FGF4 (Reliatech, Wolfenbüttel, Lower Saxony, Germany; #100-017L) and 1 μg/mL heparin (Sigma-Aldrich, St. Louis, MO, USA; #H3149-10KU).

### 4.9. Flow Cytometric Analysis

Cells were harvested with 0.05% trypsin/EDTA (Gibco, Thermo Scientific, Waltham, MA, USA; #11580626) and filtered through a 40-μm cell strainer (Becton Dickinson, Franklin Lakes, NJ, USA; #352235). After washing with fluorescence-activated cell sorting (FACS) buffer (2% fetal bovine serum (FBS) in PBS (Gibco, Thermo Scientific, Waltham, MA, USA; #11503387), cells were resuspended in FACS buffer containing 1% 7-amino-actinomycin D (AppliChem, Darmstadt, Hessen, Germany; #A7850). Detection of GFP positivity was performed using a FACSCanto (Becton Dickinson, Franklin Lakes, NJ, USA) and analyzed with FlowJo Software (TreeStar, Becton Dickinson, Franklin Lakes, NJ, USA).

### 4.10. Invasion Assay

FluoroBlok™ Cell Culture Inserts with 8 μm pore size (Corning, New York, NY, USA; #351157) were coated using 0.8 mg/mL Matrigel (Sigma-Aldrich, St. Louis, Missouri, USA) and dried for 2 h at 37 °C and then at room temperature (RT) overnight. Matrigel was rehydrated using supplemented RPMI (Gibco, Thermo Scientific, Waltham, MA, USA; #11530586) [38]. Cells were harvested using 0.05 trypsin/EDTA (Gibco, Thermo Scientific, Waltham, MA, USA; #11580626), counted and reseeded at 2 × 10^4^ cells per well, and kept under differentiating conditions (TS-Medium w/o CM, FGF4 and heparin) for five days. Medium in the lower chamber was changed every day. For analysis, Matrigel in upper chamber as well as medium in lower chamber were discarded and filters were washed several times with PBS. Filters were fixed and stained with 1% Hoechst (Sigma-Aldrich, St. Louis, MO, USA; #H6024) in methanol (VWR, Radnor, PA, USA; #20.847.307) for 10 min. After 3 washing steps with PBS, detection of invaded cells was carried out using a Leica DM LB microscope (Wetzlar, Hessen, Germany).

## Figures and Tables

**Figure 1 ijms-21-05503-f001:**
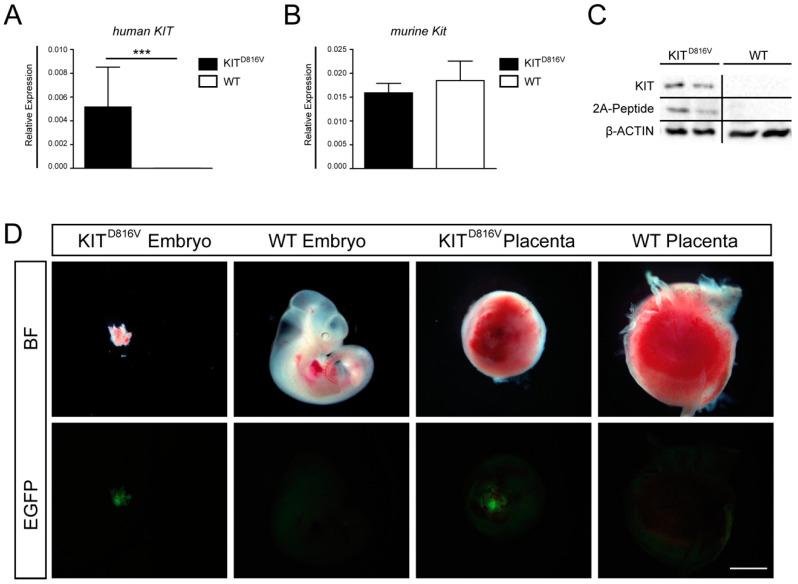
Detection of KIT^D816V^-positive embryos and placentas: (**A**) qRT-PCR for transgene expression of human *KIT* in KIT^D816V^ and wildtype (WT) placentas obtained on E10.5 and (**B**) qRT-PCR for endogenous expression of murine *Kit* in KIT^D816V^ and WT placentas obtained on E10.5. RNA was obtained from at least three biological repeats. Expression is normalized to the housekeeping gene *Gapdh*. Bars display mean value ± SD. Significance was determined by unpaired t-test and indicated with *** *p* < 0.001. (**C**) Western Blot detecting KIT and 2A-peptide in placentas at E11.5 in comparison to β-ACTIN and (**D**) photographs showing KIT^D816V^ and WT expressing embryos and placentas at E11.5 in brightfield (BF) and GFP fluorescence (scale bar: 1 mm): Experiments were performed in at least tree biological repeats.

**Figure 2 ijms-21-05503-f002:**
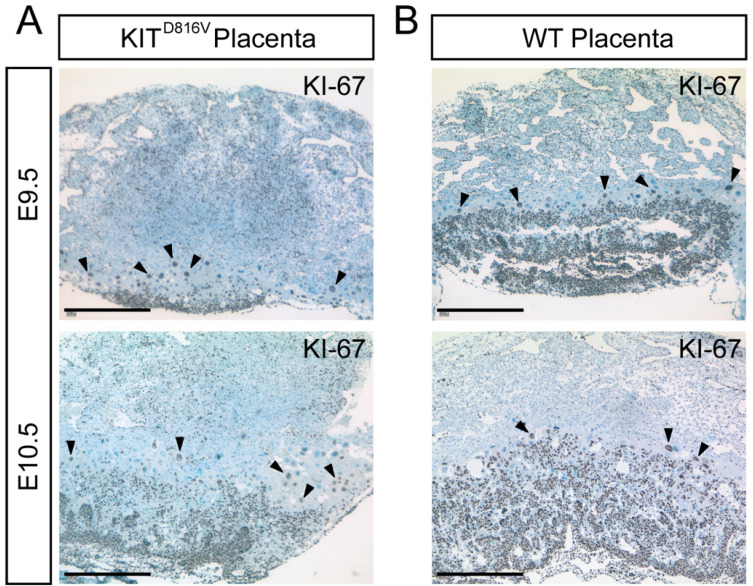
KIT^D816V^ shows a reduced number of KI-67-positive cells. (**A**,**B**) Immunohistochemical staining of paraffin sections of KIT^D816V^ and WT placentas for KI-67 at E9.5 and E10.5: Black arrowheads mark Parietal Trophoblast Giant Cells (P-TGCs) undergoing endoreduplication (biological replicates = 4; scale bar represents 500 μm).

**Figure 3 ijms-21-05503-f003:**
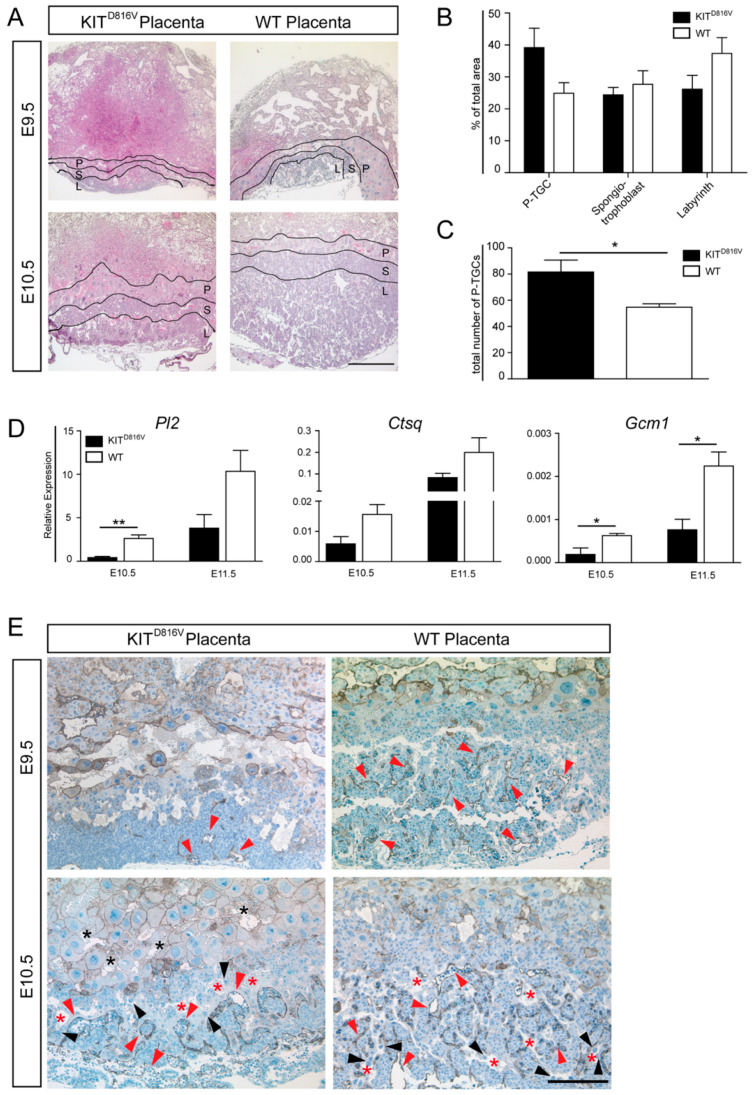
Placental structure is grossly affected by KIT^D816V^. (**A**) Photomicrographs of hematoxylin and eosin (H&E) staining of paraffin sections of KIT^D816V^ and WT placentas on E9.5 (biological replicates = 2) and E10.5 (biological replicates = 3): Black lines mark areas of the P-TGCs (P), spongiotrophoblast (S), and labyrinth layer (L). Scale bar represents 500 μm. (**B**) Quantification of P-TGC, spongiotrophoblast, and labyrinth areas at E10.5 compared to the total area of respective placenta (biological replicates = 3) and (**C**) quantification of number of P-TGCs in KIT^D816V^ and WT placenta at E10.5 using ImageJ in three biological replicates: Significance was determined by unpaired t-test and indicated with * *p* < 0.05. (**D**) qRT-PCR analysis of labyrinth-specific TGC marker *Pl2*, *Ctsq*, and *Gcm* in KIT^D816V^ and WT placentas at E10.5 and E11.5. RNA was obtained from at least three biological replicates. Expression is normalized to the housekeeping gene *Gapdh*. Bars display mean value ± SD. Significance was determined by unpaired t-test and indicated with * *p* < 0.05 and ** *p* < 0.01; (**E**) Immunohistochemical staining of CD31 on paraffin sections of KIT^D816V^ and WT placentas at E9.5 (biological replicates = 2) and E10.5 (biological replicates = 2): Red arrowheads indicate fetal endothelial cells enclosing fetal blood spaces, and black arrowheads indicate S-TGCs. Maternal sinusoids are indicated by red asterisks, and maternal blood in the P-TGC layer is indicated by black asterisks. Scale bar represents 200 μm.

**Figure 4 ijms-21-05503-f004:**
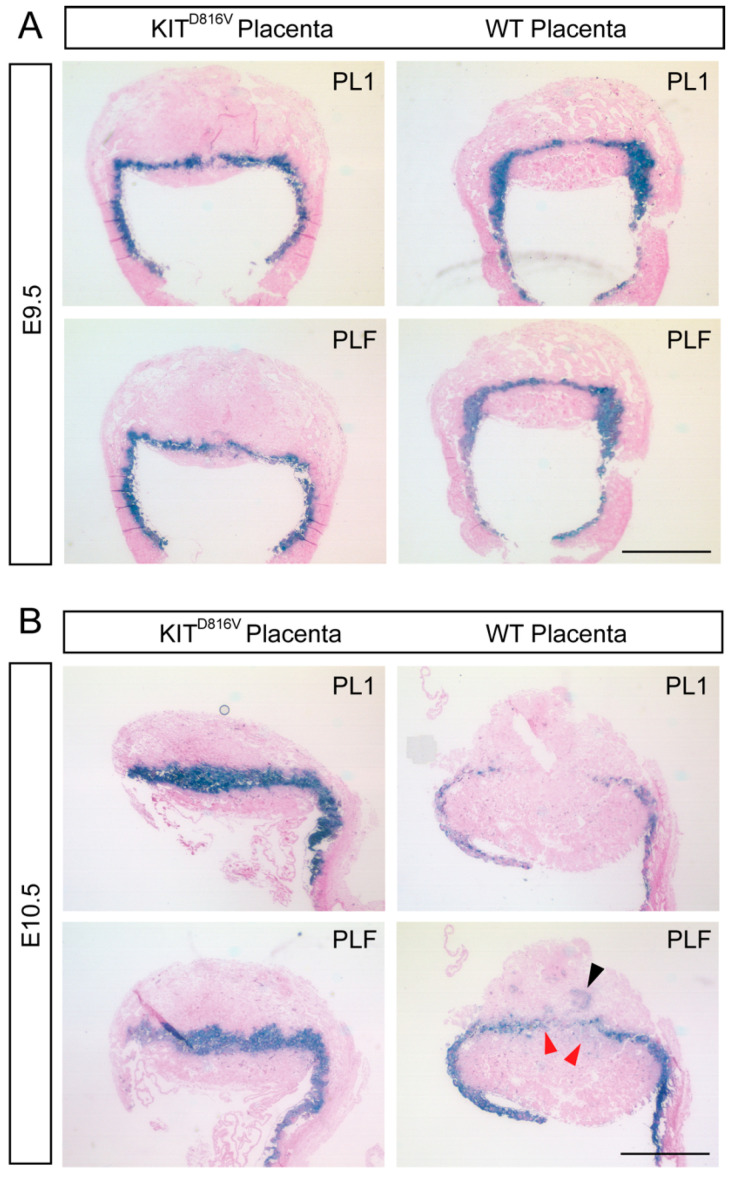
Increased levels of PL1 and PLF in KIT^D816V^ placentas: (**A**) In situ hybridization of PL1 and PLF on paraffin sections of KIT^D816V^ and control placentas at E9.5 using specific probe for PL1 and PLF (biological replicates = 2). Scale bar represents 1 mm. (**B**) In situ hybridization of PL1 and PLF on paraffin sections of KIT^D816V^ and control placentas at E10.5 (biological replicates = 2): Red arrowheads indicate PL1^−^/PLF^+^ cells. Black arrowhead indicates invading Spiral Artery (SpA-)TGCs. Scale bar represents 1 mm.

**Figure 5 ijms-21-05503-f005:**
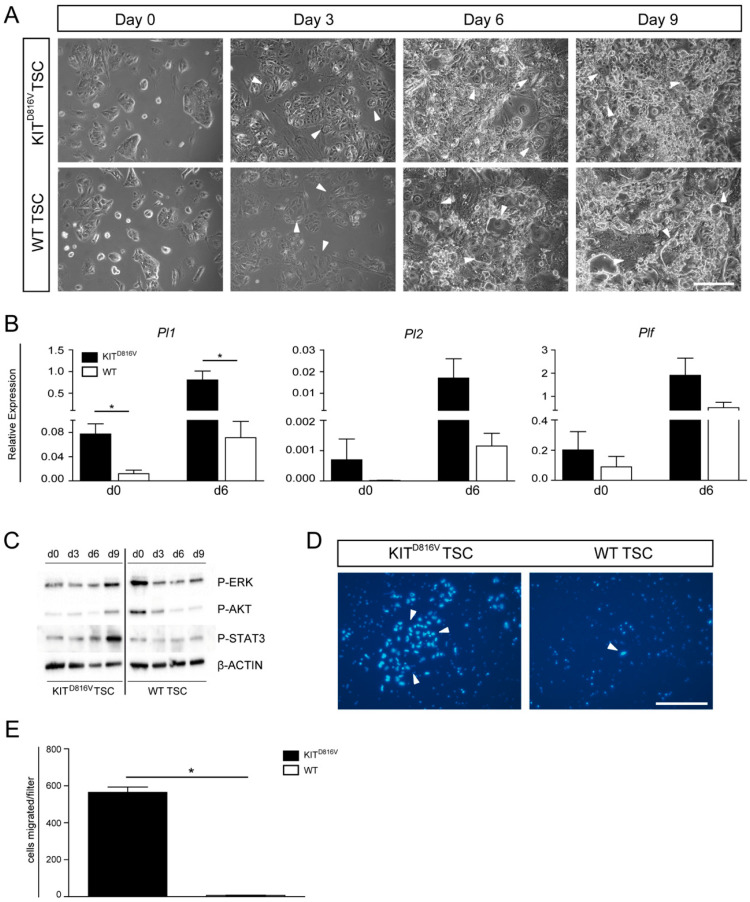
Differentiation of KIT^D816V^-TSC: (**A**) Photomicrographs depicting in vitro differentiation of KIT^D816V^-TSC line #3 and WT-TSC line 2.1 cultured for 0, 3, 6, and 9 days under differentiation conditions. White arrows indicate TGCs. Scale bar represents 250 μm. (**B**) qRT-PCR analysis of endogenous expression of *Pl1*, *Pl2*, and *Plf* in KIT^D816V^-TSC line #3 and WT-TSC line 2.1 in undifferentiated states and after culture under differentiation conditions for 6 days. RNA was obtained from three biological replicates; expression was normalized to the housekeeping gene *Gapdh.* Data is represented by mean value ± SD. Significance was determined by unpaired t-test and indicated with * *p* < 0.05. (**C**) Western Blot detected phosphorylated Extracellular-Signal Regulated Kinase (ERK), Protein Kinase B (AKT), and Signal Transducers and Activators of Transcription 3 (STAT3) during differentiation of KIT^D816V^-TSC line #3 and WT-TSC line 2.4 in comparison to β-ACTIN. (**D**) Representative photomicrographs of Hoechst-stained nuclei of cells that invaded through Matrigel coated filter membranes after five days under differentiating conditions: White arrows indicate nuclei of TGCs. Scale bar represents 250 μm. (**E**) Quantification of invaded cells per filter in KIT^D816V^-TSC line and WT-TSC line (biological replicates = 2): Data is represented by mean value ± SD. Significance was determined by unpaired t-test and indicated with * *p* < 0.05.

**Figure 6 ijms-21-05503-f006:**
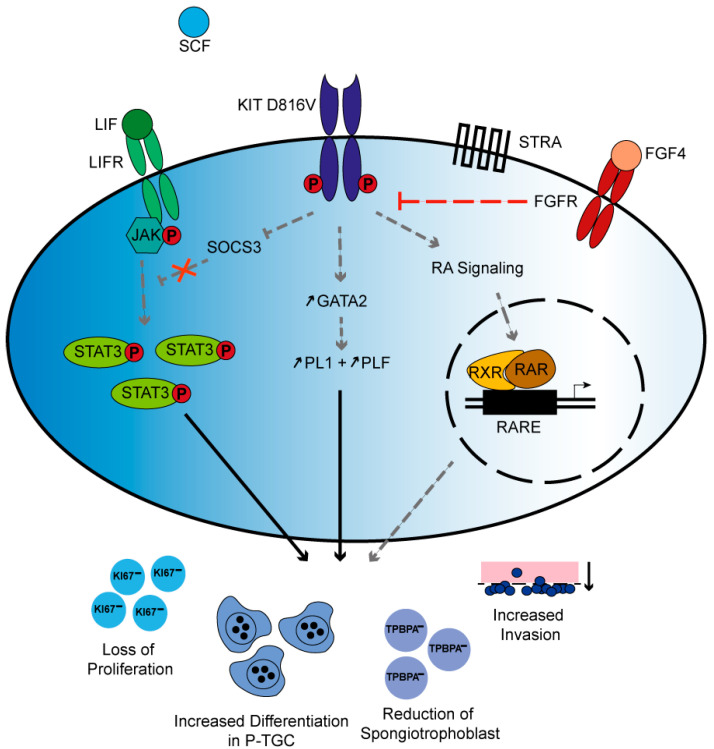
Proposed mechanism of KIT^D816V^ signaling in TSC: Schematic summary depicting possible signaling mechanisms that lead to KIT^D816V^ placental phenotype observed in this study. Independent of SCF binding, the KIT^D816V^ receptor is active. KIT inhibits SOCS3, thereby interrupting the negative feedback loop of Leukemia Inhibitory Factor (LIF) signaling resulting in accumulation of P-STAT3. Further, KIT signaling induces the expression of GATA2, which then transactivates PL1 and PLF. Independent of retinoic acid (RA) presence, KIT impinges on RA-controlled gene expression. Taken together, chronic KIT^D816V^ signaling results in severely diminished placental proliferation and reduction of spongiotrophoblast cells but increase in differentiation towards P-TGC and elevated invasion. In the presence of FGF4, Fibroblast Growth Factor Receptor (FGFR) signaling can override KIT-mediated signals. The black arrows indicate relations found in this study, whereas the gray and dashed arrows present hypothetical interactions, some of which have only been demonstrated in other cell types and T bars indicate repression.

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
