# Peer review of "Persistent Human KIT Receptor Signaling Disposes Murine Placenta to Premature Differentiation Resulting in Severely Disrupted Placental Structure and Functionality"

_ijms, 2020, doi:10.3390/ijms21155503_

Round 1

Reviewer 1 Report

The manuscript entitled "Persistent human KIT receptor signaling disposes murine placenta to premature differentiation resulting in severely disrupted placental structure and functionality" describes the influence of constitutively activated mutated KIT D816V on mouse placental development and its impact on embryonic development. It is conclusive and well written.

To indicate the importance of the KIT receptor, a paragraph about KIT knockout mice should be added.

Suggestions

Please add the correct name of the animals (this also includes the background of the strains, and whether their they are backcrossed to C57BL/6 N or J or are on a mixed background. The term non-pregnant females should be used.

Formatting notes: Follow the SI rules meaning leaving a space between number and unit except for % and °C. Be consistent with comma or dots for decimal (e.g. line 147). Leave a space between (A, B) in figure legends (e.g. line 143). Line 204: write 1 mm instead of 1000 um.

Line 104: How was the yolk sac development in the KIT mutant embryos. Was it developed normally since the embryo seems to result in absorption and the genotype may be contaminated by the mothers genotype. Was the mendelian ratio of double transgenic embryos as expected. 

Please reword in "KIT and control" to make it consistent as in the figure the KIT results are mentioned first (line 111, 119, 124, 144 …). Indicate the biological repeats (n=?) in figure legend since they appear to vary in experiments.

Figure 1: I assume that 3 embryos were dissected and 3 technical repeats were done. Please indicate how many embryos (biological repeats)  and how many technical repeats were done. qPCRs is done in triplicates and the mean was calculated from the biological repeats? is this correct. Also for other figures (n=?) as already mentioned before. Switch WT and KIT if possible in 2C to make it constant (also Fig. 5C).

The size of the WT embryo at E10.5 appears rather small. Are the embryos from different timed matings? How were the timed matings performed. 12 h in embryonic development is a long time and size changes significantly. Please add a paragraph how the timed matings were performed. Was there a sign that the CMV-cre mouse is not always recombining 100% and needs germline transmission to have 100% recombination. Please comment on that.

It is mentioned in the discussion that the Gata2 is an important target of KIT. Why was it not analyzed in this study.

Minor suggestions:

Line 188: Paragraph: please be consistent with PL1+/PLF+ (what is mentioned first and second) to make easier to read.

Figure 2, line 144: is n=3 for all three days or for each day?

Line 237: please write out TM and CM the first time it appears and put catalogue # in material and methods.

Figure 4: Please put the sequence of the probes in the supplementary tables.

Line 194: Please write out SpA-TGCs the first time it appears in the text (also for P-TGC and S-TGC).

Figure 5, Line 287: Why is the SEM shown in this image. The SD is the better/correct parameter to show the variability of the data.

Line 431: Check concentration and correct spacing of proteinase K and probe.

Line 444: Please write out TS medium and provide catalogue #. Please include for all materials and reagents the catalogue # to support reproducibily for other researchers.

Suppl. Figure S1: Please indicate the number of biological (n=?) and technical  repeats of B and C-E, 3 embryos?. What probe was used for in situ hybridisation? Please check for typos. The indication for KITD816V and WT should be in each subfigure. (B) in upper case.

Suppl. Figure S2: Please indicate the number of biological and technical repeats. Genotyping PCR is redundant, please remove. Why is no bright field image shown?

Suppl. Table , Antibodies: The pound sign means number, should read Catalogue #.

Author Response

Reviewer #1

We thank the reviewer for his detailed evaluation. We appreciate his input and hope to address all of the points appropriately.

The manuscript entitled "Persistent human KIT receptor signaling disposes murine placenta to premature differentiation resulting in severely disrupted placental structure and functionality" describes the influence of constitutively activated mutated KIT D816V on mouse placental development and its impact on embryonic development. It is conclusive and well written.

  1. To indicate the importance of the KIT receptor, a paragraph about KIT knockout mice should be added.

AUTHORS RESPONSE:

We now included a paragraph about KIT depletion in lines 75-81 reading:

“Depletion of KIT receptor results in various defects spanning from hematopoietic failure, macrocytic anemia, pigmentation deficiency and sterility to intestinal dysfunction [23–26]. Mice carrying homozygous KITW/W mutation die pre- or perinatally, however, can be rescued by microinjection of wildtype fetal liver hematopoietic cells into placentas of affected fetuses [23,24,26]. Introduction of the viable c-Kit-deficient mouse line allowed for studying loss of KIT expression in the adult mouse and showed that KIT is essential for adult lymphopoiesis in bone marrow and thymus [26].”

Suggestions

  1. Please add the correct name of the animals (this also includes the background of the strains, and whether their they are backcrossed to C57BL/6 N or J or are on a mixed background. The term non-pregnant females should be used.
  • We added the genetic background, line 506 now reads: “Both mouse strains were kept on a 129sv/S2 and C57BL/6 genetic background.”
  • Line 61: “unpregnant mice” was changed to “non-pregnant females”.
  1. Formatting notes: Follow the SI rules meaning leaving a space between number and unit except for % and °C. Be consistent with comma or dots for decimal (e.g. line 147). Leave a space between (A, B) in figure legends (e.g. line 143). Line 204: write 1 mm instead of 1000 um.
  • Spaces were introduced between number and unit.
  • Line 143 and 147 are now lines 193-195: This Figure Legend was changed completely. It now reads: “Figure 2. KITD816Vshow reduced number of KI-67 positive cells. (A, B) Immunohistochemical staining of paraffin sections of KITD816V and WT placentas for KI-67 at E9.5 and E10.5. Black arrowheads mark P-TGCs undergoing endoreduplication (biological replicates = 4; scale bar represents 500 μm).“
  • Line 204 is now line 291: 1000 μm was changed to 1 mm.
  1. Line 104: How was the yolk sac development in the KIT mutant embryos. Was it developed normally since the embryo seems to result in absorption and the genotype may be contaminated by the mothers genotype. Was the mendelian ratio of double transgenic embryos as expected. 
  • Yolk sac development in KITD816V embryos was normal. Yolk sacs could be easily distinguished, they were GFP positive and we can exclude any contamination of maternal cells. Also, the mendelian ratio was as expected.
  1. Please reword in "KIT and control" to make it consistent as in the figure the KIT results are mentioned first (line 111, 119, 124, 144 …). Indicate the biological repeats (n=?) in figure legend since they appear to vary in experiments.
  • “WT and KITD816V” was changed to “KITD816V and WT” wherever it occurred.
  • n was only missing in TPBPA staining; n means biological – not technical – repeats
  1. Figure 1: I assume that 3 embryos were dissected and 3 technical repeats were done. Please indicate how many embryos (biological repeats) and how many technical repeats were done. qPCRs is done in triplicates and the mean was calculated from the biological repeats? is this correct. Also for other figures (n=?) as already mentioned before. Switch WT and KIT if possible in 2C to make it constant (also Fig. 5C).
  • Yes, 3 embryos were dissected and at least 3 biological repeats were done.
  • Yes, the mean was calculated from three biological repeats.
  • Number of biological repeats was added in case it was missing before.
  • WT and KITD816V samples were switched in 2C and 5C.
  1. The size of the WT embryo at E10.5 appears rather small. Are the embryos from different timed matings? How were the timed matings performed. 12 h in embryonic development is a long time and size changes significantly. Please add a paragraph how the timed matings were performed. Was there a sign that the CMV-cre mouse is not always recombining 100% and needs germline transmission to have 100% recombination. Please comment on that.
  • All embryos originate from equally timed matings. The morning after mating, female mice were checked for vaginal plug. In case of plug detection, noon of that day was considered E0.5. Accordingly, mice were sacrificed 9 to 11 days later around noon. This passage is now included in 4.1 – Generation of transgenic animals (lines 507-509).
  • There was no indication of incomplete recombination.
  1. It is mentioned in the discussion that the Gata2 is an important target of KIT. Why was it not analyzed in this study.
  • We now analyzed the expression of Gata2 in KITD816V and WT TSC in an undifferentiated state and after 6 days of differentiation. Data is now included in Suppl. Fig. S2 and in lines 343-346.

 Minor suggestions:

  1. Line 188: Paragraph: please be consistent with PL1+/PLF+ (what is mentioned first and second) to make easier to read.
  • PLF+/PL1- (now in line 282) was changed to PL1-/PLF+.
  1. Figure 2, line 144: is n=3 for all three days or for each day?
  • n=3 is for each day.
  1. Line 237: please write out TM and CM the first time it appears and put catalogue # in material and methods.
  • Conditioned medium and trophoblast stem cell medium was inserted (now in line 331-332) and catalogue numbers were added in Material and Methods.
  1. Figure 4: Please put the sequence of the probes in the supplementary tables.
  • In situ hybridization was performed using protocol and DIG-labeled probes established by Simmons et al., 2007, “Diversesubtypes and developmental origins of trophoblast giant cells in the mouse placenta”. This passage is now included in 4.7 In situ hybridization - lines 572-573.
  1. Line 194: Please write out SpA-TGCs the first time it appears in the text (also for P-TGC and S-TGC).
  • SpA is now written out in line 280 where it is mentioned for the first time. P-TGC were already appropriately introduced in line 99, as well as S-TGC which were correctly introduced in line 207.
  1. Figure 5, Line 287: Why is the SEM shown in this image. The SD is the better/correct parameter to show the variability of the data.
  • SEM was changed to SD in data presentation. Updated Figure 5 is now included in the manuscript.
  1. Line 431: Check concentration and correct spacing of proteinase K and probe.
  • Concentration and spacing were corrected.
  1. Line 444: Please write out TS medium and provide catalogue #. Please include for all materials and reagents the catalogue # to support reproducibily for other researchers.
  • We wrote out TS medium and catalogue numbers are now included for all materials.
  1. Figure S1: Please indicate the number of biological (n=?) and technical repeats of B and C-E, 3 embryos?. What probe was used for in situ hybridisation? Please check for typos. The indication for KITD816V and WT should be in each subfigure. (B) in upper case.
  • Figure S1B: Number of biological repeats = 2
  • Figure S1C-E: Yes, RNA was isolated from three placentas, number of biological repeats = 3.
  • Probes for in situ hybridization: See Reviewer’s Comment 4
  • Indication is now included in each subfigure; (B) is now in upper case
  1. Figure S2: Please indicate the number of biological and technical repeats. Genotyping PCR is redundant, please remove. Why is no bright field image shown?
  • Number of biological repeats is indicated
  • Genotyping was done once for tissue (Suppl. Fig. S1A) and for derived TSC lines (Suppl. Fig S2A). We consider both PCRs to be relevant.
  • Brightfield Images of cell lines are included in Figure 6 on day 0 of differentiation.
  1. Table , Antibodies: The pound sign means number, should read Catalogue #.
  • Row title was changed to Catalogue #

Reviewer 2 Report

In this study by Kaiser et al., the authors investigate the impact of constitutively active KIT signaling in the mouse placenta. Using a transgenic mouse that expresses an active mutant form of human KIT, that is turned on by Cre recombination, they show that normal mouse placental development is severely impacted and that branching morphogenesis and development of the labyrinth layer does not occur. Using trophoblast stem cell lines derived from KITD816V-mutant blastocysts, they go on to show that while proliferation of KITD816V-TS cells appears to be normal, differentiation is not. They concluded that overexpression of human KIT appears to drive differentiation of parietal trophoblast giant cells. Unfortunately, the authors did not investigate expression of markers of labyrinth trophoblast subtypes which is curious. Admittedly, their results suggest an expansion of the junctional zone trophoblast subtypes however that does not necessarily preclude a defect in labyrinth trophoblast differentiation. Furthermore, some of the images presented in figure 2 are misleading given that similar images are presented in later figures that show different tissue morphology (see comment below). Overall, the manuscript is well-written and this is an interesting topic. However, given the inconsistencies in the presentation of the data and the lack of information related to labyrinth trophoblast subtype differentiation in TS cell experiments, I feel that this study is incomplete.

  • Data presented in figures 2 and 3 are inconsistent. In figure 2, pictures are shown of both WT and KITD816V placentas at E9.5, E10.5 and E11.5. It appears from these pictures that while the placenta is normal at E9.5, it is almost completely devoid of labyrinth and junctional zone at E10.5 and E11.5. In comparison to WT placenta sections at the gestational age, there is no evidence of Ki67 or TFAP2C expression in tissue that would be considered labyrinth or junctional zone. This result is impossible to interpret because in figure 3, pictures are shown of KITD816V placentas in which there are clearly discernible junctional zone and labyrinth. It is obvious that there is a developmental phenotype but the pictures in figure 2 are not acceptable.
  • TS cell experiments only show analysis of TGC-markers. Why were additional markers of trophoblast subtypes not included?

Author Response

Reviewer #2

We also thank this reviewer for their relevant points of criticism concerning our study. We value the remarks and addressed the concerns by performing additional experiments which are now included in the manuscript.

In this study by Kaiser et al., the authors investigate the impact of constitutively active KIT signaling in the mouse placenta. Using a transgenic mouse that expresses an active mutant form of human KIT, that is turned on by Cre recombination, they show that normal mouse placental development is severely impacted and that branching morphogenesis and development of the labyrinth layer does not occur. Using trophoblast stem cell lines derived from KITD816V-mutant blastocysts, they go on to show that while proliferation of KITD816V-TS cells appears to be normal, differentiation is not. They concluded that overexpression of human KIT appears to drive differentiation of parietal trophoblast giant cells. Unfortunately, the authors did not investigate expression of markers of labyrinth trophoblast subtypes which is curious. Admittedly, their results suggest an expansion of the junctional zone trophoblast subtypes however that does not necessarily preclude a defect in labyrinth trophoblast differentiation. Furthermore, some of the images presented in figure 2 are misleading given that similar images are presented in later figures that show different tissue morphology (see comment below). Overall, the manuscript is well-written and this is an interesting topic. However, given the inconsistencies in the presentation of the data and the lack of information related to labyrinth trophoblast subtype differentiation in TS cell experiments, I feel that this study is incomplete.

AUTHORS RESPONSE:

            The reviewer stresses a certain lack of completeness of our study. We re-did and completed several experiments in order to fill gaps mentioned by the reviewers. We would like to stress the point, that the experiments cover a broad range of approaches, starting from the initial mating of the animals leading to the standard preparation and situs images of midgestation embryo and placenta. Further, histology and immunohistochemistry is used to further detail the pathology observed. We would like to mention, that the de-novo derivation of transgenic trophoblast stem cell line is in itself a task not many laboratorys are able to perform.  We used this lines to analyze the molecular consequences in more detail. So, together with the additional experiments performed for both reviewers, we believe that the study has gained robustness and we would be delighted if the reviewer 2 could agree to our view.   

  1. Data presented in figures 2 and 3 are inconsistent. In figure 2, pictures are shown of both WT and KITD816Vplacentas at E9.5, E10.5 and E11.5. It appears from these pictures that while the placenta is normal at E9.5, it is almost completely devoid of labyrinth and junctional zone at E10.5 and E11.5. In comparison to WT placenta sections at the gestational age, there is no evidence of Ki67 or TFAP2C expression in tissue that would be considered labyrinth or junctional zone. This result is impossible to interpret because in figure 3, pictures are shown of KITD816Vplacentas in which there are clearly discernible junctional zone and labyrinth. It is obvious that there is a developmental phenotype but the pictures in figure 2 are not acceptable.

  • We redid the staining of KI-67 in KITD816V and WT placentas. Updated results can be found in Figure 2 and lines 142-152. Unfortunately, due to Covid-19 restrictions concerning order and delivery, we were not capable to perform a corresponding TFAP2C staining. Nevertheless, we feel confident of including this KI67 analysis as it clearly shows a reduced proliferation in KITD816V
  1. TS cell experiments only show analysis of TGC-markers. Why were additional markers of trophoblast subtypes not included?
  • We performed additional analysis of Tpbpa, Ctsq, Gjb3, Pcdh12, Hand1 and Tfap2c expression in KITD816V-TSC in comparison to WT-TSC under stem cell promoting and differentiation inducing conditions. Data is now included in Suppl. Figure S2. Description and interpretation of new data can be found in lines 336-351.